# Effect of Nutritional Restriction on the Hair Follicles Development and Skin Transcriptome of Chinese Merino Sheep

**DOI:** 10.3390/ani10061058

**Published:** 2020-06-19

**Authors:** Xuefeng Lv, Lei Chen, Sangang He, Chenxi Liu, Bin Han, Zhilong Liu, Mayila Yusupu, Hugh Blair, Paul Kenyon, Stephen Morris, Wenrong Li, Mingjun Liu

**Affiliations:** 1College of Life Science and Technology, Xinjiang University, Urmuqi 830000, China; lvxuefeng@xjaas.net; 2Institute of Animal Husbandry Quality Standards, Xinjiang Academy of Animal Sciences, Urmuqi 830000, China; 3Key Laboratory of Genetics, Breeding & Reproduction of Grass-Feeding Livestock, Ministry of Agriculture, Urmuqi 830000, China; chenlei0991@126.com (L.C.); Hesangang3@163.com (S.H.); chenxi_4000@163.com (C.L.); hanbin0991@163.com (B.H.); lxfxjxm@outlook.com (Z.L.); mayila09910@163.com (M.Y.); 4Key Laboratory of Animal Biotechnology of Xinjiang Institute of Animal Biotechnology, Xinjiang Academy of Animal Science, Urmuqi 830000, China; 5International Sheep Research Centre, School of Agriculture and Environment, Massey University, Palmerston North 4442, New Zealand; h.blair@massey.ac.nz (H.B.); p.r.kenyon@massey.ac.nz (P.K.); s.t.morris@massey.ac.nz (S.M.)

**Keywords:** Chinese Merino sheep, wool follicle, nutritional restriction, transcriptome

## Abstract

**Simple Summary:**

The morphogenesis hair follicles begins in the embryonic stage. The first follicles formed are the primary wool follicles (PF), followed by secondary wool follicles (SF), and then secondary-derived follicles that branch from the SF in Merino sheep. The morphogenesis and development of SF in sheep determine wool yield and quality. At present, the mechanism of SF development and branching is still unclear. In this study, hair follicle morphogenesis and development were inhibited through nutritional restrictions during pregnancy, and transcriptome sequencing was performed to screen for differentially expressed genes related to the SF development in the fetus. Our findings are helpful to understand the mechanisms of SF development and branching of Merino sheep.

**Abstract:**

The high concentration of secondary branched wool follicles is a distinctive feature of the Merino sheep. At present, the molecular control of the development and branching of secondary wool follicles (SF) remains elusive. To reveal the potential genes associated with the development of hair follicles, we investigated the characteristics of prenatal and postnatal development of wool follicles, and the transcriptional expression profile in fetuses/lambs from dams under either maternal maintenance or sub-maintenance (75% maintenance) nutrition. The density of SF and the ratio of SF to primary wool follicles (PF) were reduced (*p* < 0.05) in fetuses from day 105 to 135 of gestation under sub-maintenance nutrition. Differentially expressed genes were enriched in the binding, single-organism process, cellular process, cell and cell part Gene Ontology (GO) functional categories and metabolism, apoptosis, and ribosome pathways. Four candidate genes, *SFRP4*, *PITX1*, *BAMBI*, and *KRT16*, which were involved in secondary wool follicles branching and development, were identified. Our results indicate that nutritional intervention imposed on pregnant ewes by short-term sub-maintenance nutrition could provide a strategy for the study of wool follicle development. Overall insight into the global gene expression associated with SF development can be used to investigate the underlying mechanisms of SF branching in Merino sheep.

## 1. Introduction

In sheep, wool follicle morphogenesis begins in the embryonic stage and requires coordination and communication of multiple signals to regulate their growth [1]. The Chinese Merino sheep was developed in 1985 and is characterized by its large body size, high density, and white wool with a fineness (diameter) between 18 and 25 µm [2]. Wool yield is the most important economic driver for fine wool sheep, which is affected by many factors [3]. The density of wool follicles affects wool yield while the diameter determines the wool’s value. In follicle development, the first follicle formed is the primary wool follicle (PF) which occurs at approximately day 70 of gestation, followed by secondary wool follicles (SF), starting at approximately day 85 of gestation. The secondary-derived wool follicles begin to be initiated at approximately day 105 of gestation [1]. Based on previous studies, nutrition is crucial for SF morphogenesis and development, and malnutrition directly affects the density of wool follicles [4]. Moderate nutritional restriction in ewes during days 85–135 of gestation affects the morphogenesis and branching of SF, resulting in slower SF maturation and a reduced proportion of secondary to primary wool follicles (S/P) [5].

At present, many candidate genes related with economically important traits of sheep have been discussed [6]. However, the genes related to wool quality and yield still need further study. The morphogenesis and development of wool follicles involve a complex process that are dependent on a number of genes and signaling pathways [7]. The morphogenesis of wool follicles in Merino sheep is similar to those of cashmere goats [8]. Studies on wool follicle cycling and SF development in cashmere goats have identified a large number of genes related to follicle and fiber development in which the mechanism of SF development has been elucidated using high throughput RNA sequencing approach [9,10,11,12]. However, the morphogenesis and genes or signaling pathways that regulate fetal SF development and branching in Merino sheep require further investigation. To study the spatial-temporal complexity of follicle development, we collected fetal and lamb skin tissue at five different time points during pregnancy under the conditions of regular feeding and restricted feeding. This experimental design aimed to investigate the PF and SF formation and its characteristics and to obtain global gene expression profiles of differentially expressed genes (DEGs). The purpose of this study was to identify possible genes involved in the development of SF and secondary-derived wool follicles and to evaluate the effects of sub-nutrition at different gestational moments at wool follicle and development. This knowledge can be used to understand the molecular mechanisms involved in SF branching in fine wool sheep.

## 2. Material and Methods

### 2.1. Study Design

Chinese Merino ewes weighing 40–60 kg with wool fiber thinness between 20 and 23 µm were selected from a breeding farm in Urumqi, located in the Xinjiang Province of China. Estrous cycles were synchronized using vaginal medroxyprogesterone acetate-impregnated sponges, which were removed 48 h before artificial insemination. Semen from 3 different Chinese Merino rams with wool fiber diameter 20–23 µm were used for artificial insemination. Thirty days after artificial insemination, ewe pregnancy was detected by a B-ultrasound scan. Based on previous studies of hair follicles morphogenesis and development [1,13], the effects of nutrition in five time periods were examined. These included days 55 to 85, 85 to 105, and 105 to 135 of gestation, day 135 of gestation to 7 days after birth, and 7 to 35 days after birth. During each period ewes were offered either sub-maintenance or maintenance feeding with 3 ewes in each subgroup (Figure 1). The study design included a total of 30 ewes and the number of groups are shown in Figure 1. The impacts of dam postnatal nutrition were also examined, because it is known that secondary wool follicle development and maturation can take up to 5 weeks after birth [14] and the lamb is dependent on its dams milk during that period.

### 2.2. Animal Feeding

All ewes were subjected to identical feeding regimens. The nutritional requirements for M group were based on the feeding standard of the New Zealand Society of Animal Production (NZSAP) (Appendix A). Ewes were fed a mixed pellet that contained 8.92 MJ metabolizable energy per kg. The nutrient composition of the pellet consisted of 94% dry matter, 13.55% crude protein, 2.25% crude fat, 28.41% neutral cellulase, 8.89% acid cellulase, 6.26% crude ash, 0.48% calcium, and 0.49% phosphorus. Daily feed intake of Sub-M group ewes was 75% of the M group. The daily feed intake in each group is shown in Appendix A. The animals used in this study were managed at the Research Base of Sheep Breeding of Xinjiang Academy of Animal Science, located at Urumqi city in Xinjiang province of China. Surgeries were performed under strict aseptic conditions and to minimize potential animal welfare effects. All animal handling procedures were carried out in strict accordance with approved guidelines of the Institutional Animal Care of Xinjiang Academy of Animal Science (No:20180603001).

### 2.3. Biochemical Examination of Blood

To determine the effect of sub-maintenance nutrition on the ewes and fetal development, ewe blood biochemical indexes were assayed every 7 days after start of each nutritional restriction. Blood samples were collected from the jugular vein and placed in centrifuge tubes. Centrifugation was performed at 3000 rpm for 10 min and the serum was collected and analyzed by an automated biochemistry analyzer (Olympus AU 400, Olympus-diagnostic, Hamburg, Germany). The analyzed parameters included the following: Total protein (TP), Albumin (ALB), Alanine aminotransferase (ALT), Aspartate aminotransferase (AST), urea nitrogen (BUN), blood glucose (GLU), triglyceride (TG), high-density lipoprotein (HDLC), low-density lipoprotein (LDLC), Free fatty acids (FFA). 

### 2.4. Histological Analysis of Wool Follicles

At the end of each feed restriction period, fetuses were removed by caesarean section. The body length, chest circumference, abdominal circumference, body weight, and the weight of internal organs were measured. Fetus skin samples were collected from the right mid-side of the fetuses, and then placed in centrifuge tubes containing 4% paraformaldehyde solution. Alcohol concentrations of 50%, 75%, 85%, 95%, and 100% were used separately for 1 h tissue dehydration, followed by tissue clearing for 2 h using xylene and then submerging the samples in 56 °C wax for 6 h. The samples were then embedded in paraffin at 60 °C and cut to 5 μm diameters using a micro-slicing machine (Leica RM2235, Leica Biosystems Ltd., Shanghai, China). The tissue sections were stained using Hematoxylin-eosin (HE), fixed with neutral resin, and photographed and observed under microscope. Primary and secondary wool follicle density and the S/P were measured. The PF and SF were differentiated through the presence/absence of sweat and sebaceous glands. Three independent slices were measured for each sample, with at least 10 different microscopic fields examined for each slice. The data were expressed the mean ± standard deviations (SD). A Tukey’s test following one-way ANOVA in SPSS 19.0 (IBM, NY, USA) was used for a significance analysis.

### 2.5. Transcriptome Sequencing

Skin samples were collected from all treatment groups and immediately frozen in liquid nitrogen. Following histological analysis of wool follicles, a significant difference in SF and S/P between M and Sub-M groups at days 135 of gestation were identified. Hence, 3 independent skin samples from day105 of gestation for the M group (D105), day135 of gestation for the M group (D135) and D135 of gestation for the Sub-M group (DR135) were analyzed. Total RNA was isolated using Trizol Reagent (Invitrogen, CA, USA), according to the manufacturer’s instructions. RNA libraries for each skin sample were constructed. Oligo (dTs) were used to isolate poly (A) mRNA. The mRNA was fragmented and reversed transcribed using the random primers. Second-strand cDNAs were synthesized using RNase H and DNA polymerase I. Double-strand cDNAs were then purified using a QIAQuick PCR purification kit (QIAGEN, Dusseldorf, Germany). The expected fragments were purified via agarose gel-electrophoresis after PCR amplification. Finally, amplified fragments were sequenced using the Illumina HiSeqTM 2000 instrument (Novogene, Beijing, China), according to the manufacturer’s specification.

### 2.6. Data Processing

Raw reads of all samples were pre-processed via the removal of adaptors with more than 5% unknown nucleotides. Low-quality reads, which are the percentage of low-quality bases, were also removed. Clean reads were aligned against the reference genes Ovis aries (Oar_v3.1) with SOAPaligner/SOAP2 and were annotated. Mismatch of no more than two bases was allowed for alignment to obtain uniquely mapped reads. The fragments per kilobase of exon model per million reads mapped (FPKM) was used to normalize the transcripts and estimate the expressed gene levels. Differentially expressed genes (DEGs) were identified between the two groups using the DESeq2 R package (version 1.8.1) (https://bioconductor.org/packages/release/bioc/vignettes/DESeq2/inst/doc/DESeq2.html). The Benjamini and Hochberg false discovery rate (FDR) was applied for *p*-value correction in multiple hypotheses. In this study, an adjusted *p*-value(padj) of < 0.05 and |log2 (fold change)| > 0.5 were set as the criteria to filter the DEGs.

### 2.7. GO and KEGG Pathway Analysis

GO (gene ontology) annotation was analyzed using Blast2GO software (version 2.3.5, Biobam, Valencia, Spain) (https://www.geneontology.org). The enrichment analysis of KEGG (Kyoto Encyclopedia of Genes and Genomes) pathways was implemented using KOBAS (version 3.0) (http://kobas.cbi.pku.edu.cn/index.php), padj < 0.05 were considered significantly enriched. The gene interaction network was drawn using Ingenuity Pathway Analysis (IPA: http://www.ingenuity.com) software.

### 2.8. Validation of Transcriptome Data Using qRT-PCR

To validate RNA-Seq sequencing data, 8 DEGs were selected for qRT-PCR analysis. Primers used in the qRT-PCR were designed using primer 5 (Appendix A). The PCR conditions were carried out as follows: initial denaturation at 95 °C for 10 min, followed by 40 cycles of 5 s denaturationat 95 °C, annealing at 60 °C for 30 s and extension at 72 °C for 20 s. The qRT-PCR reaction of each gene in each sample was repeated three times, the relative expression of each gene was normalized to GAPDH and calculated using the 2^−ΔΔCt^ method, and final results are shown as means ± SD.

## 3. Results

### 3.1. Changes in Blood Biochemical Indexes

Most blood biochemical indexes showed no difference between pregnant ewes under M and Sub-M nutrition. However, HDLC was higher (*p* < 0.05) in M than in Sub-M on days 14, 21, and 28. Blood glucose level was lower (*p* < 0.05) for M compared to Sub-M treatments on day 21. NEFA was lower (*p* < 0.05) in M than Sub-M treatments on day 28. No other differences were observed (*p* > 0.05) (Appendix A).

### 3.2. Morphogenesis and Development of Wool Follicles

There was no difference in fetus weight, body size, and organ development between nutritional groups. However, kidney weight was significantly (*p* < 0.05) lower in the Sub-M treatment compared to the M group at day 135 of gestation (Appendix A). On day 85 of gestation, PF with a hairy bud-like structure surrounded by sebaceous glands were observed (Figure 2). The arrangement of wool follicles appeared regular and the morphology of SF featured an aggregation of dermal cells into clumps. On day 105 of gestation, dicotyledonous sebaceous glands were found around the PF, while the SF bud had extended downward and had penetrated deep into the dermis. With the increase of the diameter of the dermal papillae, the sebaceous glands began to form, although the hair shafts were not observed in PF and SF at this stage. On day 135 of gestation, completed sebaceous and sweat glands were formed around the PF. Sebaceous glands with bilobed structures were prominent and sweat glands were visible in lumen structures. SF had no sweat glands but contained sebaceous glands. In contrast to SF, the PF grew some hair shafts. On day 7 after birth, SF were still growing with some of them being devoid of hair shafts. In both M and Sub-M groups, most of the wool follicles grew hair shafts by 35 days after birth (Figure 2).

### 3.3. The Density and S/P Ratios of Wool Follicles

The density of PF, SF, and S/P ratios between M samples and Sub-M samples at days 85 and 105 of gestation showed no significant differences (*p* > 0.05). The SF density, total density and S/P ratio of Sub-M samples were lower (*p* < 0.05) than those of the M group at days 135 of gestation. There was no difference (*p* > 0.05) in the densities of PF and SF, or S/P ratios between the M and Sub-M at 7 days after birth. The S/P ratios of M samples were higher (*p* > 0.05) than those of the Sub-M at day 35 after birth (Table 1).

### 3.4. DEGs Analysis

The transcriptional profiles of D105, D135, and DR135 samples revealed a total of 20,062, 20,653, and 20,846 transcripts, respectively. A total of 2164, 2739, and 692 DEGs in D135 vs. D105, DR135 vs. D105, and DR135 vs. D135, respectively, were obtained (Appendix A). Co-expressed DEGs, specific DEGs and comparisons of D135 vs. D105, DR135 vs. D105, and DR135 vs. D135 are shown in Figure 3a. Up-regulated and down-regulated genes for comparisons between D135 vs. D105, DR135 vs. D105 and DR135 vs. D135 are shown in Figure 3b. Nutritional restriction was associated with changes in the number of DEGs.

### 3.5. GO and KEGG Analysis of the DEGs

To understand the processes driving the changes in histomorphology of wool follicles after feed restrictions, GO terms analysis of DEGs was carried out. The DEGs of D135 vs. D105, DR135 vs. D105, and DR135 vs. D135 were annotated in 54, 54, and 46 functional categories, respectively (Figure 4). For biological processes, the top three functional categories were single-organism process, cellular process, and metabolic process, all three are involved in cell proliferation and tissue or organ development. It also includes most of the known genes related to wool follicle development. In terms of cellular components, the top three functional categories were cell, cell part, and organelle, these include genes that are related to cell structure, junction, and communication. For molecular function, most genes were classified into binding, which mainly involves protein binding, ion binding, carbohydrate derivative binding, and sulfide binding-related genes.

DEGs were also analyzed by KEGG enrichment pathway. In D135 vs. D105, Protein digestion and absorption, tight junction and cell adhesion molecules (CAMs) pathways were the most abundant pathways (padj < 0.05) in the top 20 KEGG pathways (Figure 5a). Many genes related to hair follicle development, including *CAV3, COL6A*, *COL21A*, *ITGA9, VDR*, *MAP3K7*, *TGF-β2*, *PLCB, WNT2*, and *WNT16*, were identified in these pathways. In DR135 vs. D105, apoptosis and Mucin type O-glycan biosynthesis were the most abundant pathways in the top 20 KEGG pathways (Figure 5b). In addition, the pathways related to hair follicle development, including cell adhesion molecules (CAMs), ECM-receptor interaction and tight junction, were also found in the top 20 pathways. In DR135 vs. D135, ribosome and oxidative phosphorylation were the most significantly enriched pathways (Figure 5c).

### 3.6. Changes in Signaling Pathway Genes Related to Wool Follicle Development

The Wnt and TGF-β/BMP signaling pathways are closely related to wool follicle development. To understand the changes in genes related to these two signaling pathways, key genes for wool follicle development were screened. A total of 20 Wnt signaling pathway-related genes and 23 TGF-β/BMP signaling pathway-related genes were identified among DEGs. The expression patterns of the two signaling pathway-related genes for D105, D135, and DR135 samples were analyzed using a cluster heat map. Among the 20 Wnt pathway-related genes, 12 genes were down-regulated and 8 were up-regulated from day 105 to 135 of gestation in M samples (Figure 6a). The expressions of *WNT2*, *WNT5A*, and *WNT16* were down regulated at D135 when compared to the D105. *DKK* and *SFRP* are inhibitors of Wnt signaling. The analyses of these inhibitors in D135 showed that the expressions of *DKK1*, *SFRP2*, and *SFRP4* were down-regulated, whereas *DKK2*, *SFRP1*, and *SFRP5* were up-regulated. In the Sub-M group (DR135), the gene expression profile mostly displayed a similar trend to the D135 group. However, the expression of *FRAT2* was down-regulated in D135 but up-regulated in DR135. The expression of *PLCB4* and *SFRP1* showed an up-regulated trend in D135 and a down-regulated trend in DR135 in comparison with D105.

Among the TGF-beta/BMP pathway-related genes, 12 genes were down-regulated and 11 were up-regulated from days 105 to 135 of gestation in M group (Figure 6b). The expressions of *MYC, BMP2*, *BMP4* and *BAMBI* were significantly up-regulated, while the expression of *IGF1, PITX2, BMP5*, *NOG*, *TGF-β2*, *BMP3* and *TGF-β3* were significantly down-regulated. Compared with D135, the expression of these genes was further up-regulated or down-regulated in DR135. The expression of the *BAMBI* was greatly up regulated at DR135 vs. D135. The expression of the *IGFBP3* was up-regulated at D135 vs. D105 and down-regulated at DR135 vs. D105.

### 3.7. Identification of Key Genes for Secondary Follicles Development and Branching

The branching of secondary follicles (SF) in fine wool sheep occurs between 105 and 135 days of gestation. SF emerge in the neck of developing secondary original follicles that leads to the formation of secondary-derived follicles. The functional genes involved in this process and their mechanisms are currently unclear. To further investigate the mechanism of secondary follicle development in fine-wool sheep, genes related to wool follicle development, epithelial cell proliferation and regulation, keratin differentiation, and branching morphology were screened for DEGs. (Table 2). We found significant differences in the expression of *BMP5*, *CAV3*, *DNASE1L2*, *FOXN1*, *FOXA1*, *SLURP1*, *MMP12*, *SFRP2*, *SFPR4*, *PITX1*, *PITX2*, *KRT16, BAMBI*, and *SIX4* from 105 to 135 days of gestation. Meanwhile, the expression of *SFRP4*, *PITX1*, and *KRT16* was significantly different in DR135 vs. D135, wherein the expression of *PITX1* was up-regulated and the expressions of *SFRP4* and *KRT16* were down-regulated. These findings indicate that the changes in expression levels may be related to the development and branching of SF.

### 3.8. Validation of DEGs by qRT-PCR

To evaluate the validity of RNA-seq data, eight transcripts were selected for examination by qRT-PCR. The results showed that the expression patterns both qRT-PCR and transcriptome sequencing data were consistent for 8 genes (Appendix A), which demonstrated the reliability of transcription data.

## 4. Discussion

### 4.1. Wool Follicle Morphogenesis and Development

According to previous reports [15,16], the appearance of PF in fine wool sheep begins at about 40 days of fetal life, whereas the SF appear from around 90 to 126 days. In this study, the PF appeared in the fetus at 55 day of gestation while the SF appeared at 85 day, both being similar to the previous studies. The S/P ratio increased rapidly between 105 and 135 days of gestation, revealing that this was a crucial period for SF development. Sub-M nutrition of the dam partially inhibited the development of SF and reduced the number of SF. The development of PF and SF densities were similar between the Sub-M and M groups. As the fetus gained weight, the skin area also increased and consequently the density of PF gradually decreased. PF densities decreased rapidly during days 105 to 135 of gestation. By contrast, the density of SF increased during this period, and in particular, the density of SF was the highest at 135 days of gestation in the M group. In contrast to the density parameter, the S/P ratio increased rapidly in late pregnancy in both treatment groups but became relatively stable after birth.

Nutrition of pregnant ewes has the potential to affect the establishment of the wool follicle population in the fetus [5]. In this study, there was no difference in the density of PF between the M and Sub-M groups, indicating that Sub-M nutrition did not affect the morphogenesis and development of PF. This was consistent with the result of Schinckel and Short [17] who showed that PF density was independent of nutritional status of the fetus. There was no difference in SF density between Sub-M and M groups at days 85 and 105 of gestation. It is possible that the level of sub-nutrition was not sufficient to change wool follicle development. Between days 105 and 135 of gestation, SF branching occurred and wool follicle density in the Sub-M group was lower than that of the M group at day135 of gestation. This reduction may be due to the decrease and delay of SF branching caused by lower dam nutrition, which also resulted in a decrease in the S/P ratio, which is consistent with previous findings [18,19]. In our study, the S/P ratio of the Sub-M group was 13% lower than that of M group, which is consistent with Schinckel and Short [16]. However, there was no difference in S/P ratio after birth, suggesting that changes in SF population were delayed in Sub-M group. We also found that more SF did not grow hair shafts in the Sub-M group from day 135 of gestation to7 days after birth when compared to the M group. The maturation of SF is completed by approximately 12 weeks after birth [15]. Thus, it is likely the Sub-M treatment inhibited the development of wool follicles and delayed the development of the hair shafts.

Overall, our research showed that Sub-M nutrition led to a reduction in SF density by inhibiting the branching of SF during days 105 to 135 of gestation and provided an experimental model to study the molecular mechanisms of the formation of SF.

### 4.2. Identification of Genes Related to Secondary Hair Follicle Branching

We found that Sub-M nutrition hindered the development and branching of SF in fetuses alongside several changes in gene expression profiles. Menzies [20] reported that *FGF10* and *BMP4* were associated with secondary wool follicle branching in sheep. However, the genes related to secondary wool follicle branching are still uncertain. In this study, we provide an overall insight into global gene expression associated with Merino sheep wool follicle development and branching.

Wnt and TGF-β/BMP signaling pathways play an important role in regulating wool follicles development [21,22,23,24]. We identified several genes related to the Wnt and TGF-β/BMP signaling pathways in DEGs, such as *WNT2*, *SFRP2*, *SFRP4*, *BAMBI*, *BMP2*, and *BMP5*, which play an important role in hair follicle development. Using functional annotation and differential expression, we found there were differences in the expression of *KRT16*, *NDASE1l2*, *CAV3*, and *PITX1* in DR135 and D135 fetuses. Using the same analyses, we identified 4 genes (*SFRP4*, *PITX1, BAMBI*, and *KRT16*) that might be essential in SF development and branching in Merino sheep follicles.

During the period of PF maturation, which is characterized by SF branching and hair shaft formation from 105d to 135 d of gestation, both activation and inhibition of genes occurs in the Wnt and TGF-β/BMP signaling pathways [25]. In contrast to D105, the expression at D135 of *NOG* was down-regulated and *BAMBI* up-regulated in TGF-β/BMP signaling pathway. Meanwhile, in the Wnt signaling pathway, the expressions of *DKK2*, *SFRP1*, and *SFRP5* were up-regulated, while the expressions of *DKK1*, *SFRP2*, and *SFRP4* were down-regulated at D135 vs. D105. These results suggest that the activation and inhibition modes of Wnt signaling occurred during wool follicle development. Inhibitors are involved in coordinating a regulatory balance during wool follicle development. The *SFRP* family seemed to play a more important role than *DKK* in wool follicle development during days 105–135 of gestation. *SFRP4* inhibits the Wnt signaling pathway by phosphorylating β-catenin, thereby causing an arrest of cell division during the G2/M phase. *SFRP4* inhibits wool follicle regeneration [26,27,28], and exogenous *SFRP4* inhibits the proliferation of cultured human keratinocytes and promotes the apoptosis and differentiation of human keratinocytes in vitro [29,30]. The *SFRP* family increased Wnt activity at low concentrations and reduced Wnt activity at high concentrations [31]. The expression of *SFRP4* was significantly down regulated in DR135 compared with D135. Here, we hypothesize that *SFRP4* selectively inhibits different Wnt ligands and frizzled receptors, thus mediating the Wnt signaling pathway, and making it responsible for the decrease of secondary follicle branching and S/P ratio under dam Sub-M nutrition. However, the mechanism in regulating wool follicle development requires further investigation.

*PITX1* belongs to the paired-bicoid homeobox protein family. It regulates embryonic development and differentiation and promotes cartilage and myogenesis in the hind limbs of mammals [32]. It also inhibits cell proliferation and migration and induces apoptosis [33,34]. *SOX9* is a direct transcriptional target gene of *PITX1.* Vidal et al. [35] showed that *SOX9*, as one of the markers of stem cells, was expressed in the outer root sheath and bulge region of wool follicles. Overexpression of *SOX9* can improve the cloning and proliferation of wool follicle outer root sheath cells [36]. Differentiation of stem cells may be necessary for SF branching. In this study, the expression of *PITX1* was up regulated in DR135 compared to D135 samples, which may play an important role in keratinogenesis inhibition and wool follicle stem cell differentiation. It is also possible that these genes are related to SF branching.

TGF-β/BMP is an indispensable signaling pathway in wool follicle development. *TGF-β2* regulates wool follicle development and growth cycles [37,38,39].The formation and development of wool follicles were inhibited, and the number of wool follicles was decreased in *TGF-β2* knockout mice [40]. *BAMBI* is an inhibitor of the *TGF-β* family. It has a similar protein sequence to the *TGF-β* I receptor and can bind to type II receptors to form a ligand-receptor complex. Due to *BAMBI* lacking the serine/threonine kinase domain in the intracellular domain, it cannot be phosphorylated, consequently, it blocks the signal transduction of the *TGF-β* family [41]. In our data, the expression of *BAMBI* was up-regulated, whereas the expression *of TGF-β2* and *TGF-β3* were down-regulated in DR135 vs. D135. It is suggested that *BAMBI* may affect SF development through the *TGF-β* pathway.

Keratin (*KRTs*) and Keratin-related protein (*KRTAPs*) are the most important component of wool fiber because they determine the structural characteristics of wool; they also play an important role in the development of wool follicles and wool fiber traits [42,43,44,45]. *KRTs* and *KRTAPs* gene expression profiles change with keratinocyte differentiation. The progression of hair follicle differentiation is characterized by the sequential activation of distinct sets of hair-specific *KRTs* and *KRTAPs* genes [46]. From 105 to 135 days of gestation, the expression of most *KRTs* and *KRTAPs* were up-regulated, except for four down-regulated *KRTs*, suggesting that the proliferation of hair keratinocytes was significantly active after 105 days of gestation. After feed restriction, seven *KRTs* and *KRTAPs* genes were down regulated. These results indicate that feed restriction can inhibit the proliferation of keratinocytes. Day 105–135 of gestation is the development and branching stage of SF in Merino sheep. The changes in *KRTs* and *KRTAPs* gene expressions were closely related to the development of SF. Expression of *KRT16* was up-regulated in D135 vs. D105 but was down-regulated in DR135 vs. D105. We propose that *KRT16* plays an important role in the development and branching of SF.

To determine the regulatory pathways of these 4 genes, we drafted the gene interaction networks using the IPA software (Figure 7). We found that *BMP2*, *BMP4*, *SOX9*, *TGF-β2,3*, *MYC*, and *CCND1* were in key positions of the networks. Of note, the *SFRP4* and *PITX1* are located upstream of these genes. *SFRP4* inhibits the Wnt signaling pathway by phosphorylating β-catenin, thereby blocking Wnt pathway transduction and regulating the transcription of target genes *CCND1* and *MYC*. Wnt signaling is necessary for primary follicle morphogenesis and can promote the proliferation and differentiation of wool follicles stem cells. *PITX1* is the downstream target gene of Wnt signaling pathway. On one hand, *PITX1* regulates cell proliferation via the RAS pathway [47], while on the other hand, it regulates the differentiation of wool follicle stem cells by acting on *FOXA1* and *SOX9*. *SOX9* is essential for wool follicle regeneration. *BAMBI* is an inhibitor of the members of the TGF-*β* family and plays a role in wool follicle development. *KRT16* is related to keratinocyte proliferation and also interacts with TGF-*β*. Hence, these four genes were closely related to wool follicle development and their functions are related to cell proliferation. In combination with the biological events of SF development during days 105 to 135 of gestation, we speculate that these genes play an important role in SF branching through the Wnt and TGF-*β* pathways.

Secondary wool follicle development and branching are complex processes in Merino sheep. They involve many signaling molecules and regulatory factors that cooperate with each other. The Wnt and TGF-β/BMP signaling pathways seem to be very important for the branching of SF, with both activation and inhibition coexisting to regulate the development of wool follicles. *SFRP4*, *PITX1, BAMBI*, and *KRT16* warrant further investigation for their roles in the development of SF in Merino sheep.

## 5. Conclusions

Sub-M nutrition (75% maintenance nutrition) has no significant effect on either ewe production and health or fetal development. There was no difference in the morphogenesis and development of fetal PF between the M and sub-M groups. However, the development and branching of SF were affected, such that SF density and S/P ratio were reduced following sub-maintenance feeding. Examination of the prenatal and postnatal development of wool follicles under pregnancy maintenance feeding and sub-maintenance nutrition in Merino sheep demonstrated that Wnt and TGF-β/BMP signaling pathways play a crucial role in wool follicle development during days 105 to 135 of gestation. Further, *SFRP4*, *PITX1*, *BAMBI*, and *KRT16* may be involved in SF branching and development in Merino sheep.

## Figures and Tables

**Figure 1 animals-10-01058-f001:**
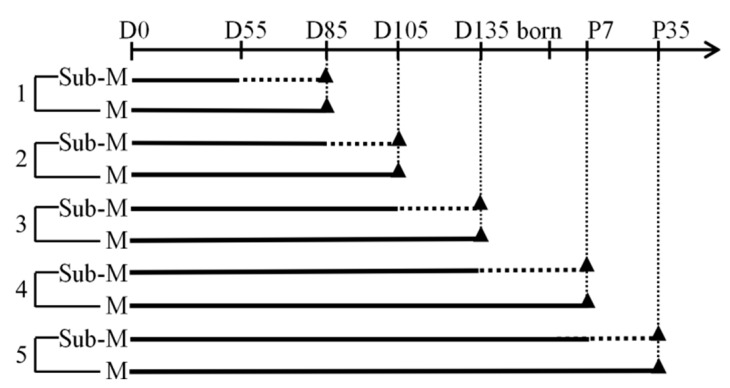
Timelines of the maternal feed restriction experiment. The experiments were divided into five groups (1–5), each group included sub-maintenance (Sub-M) nutrition ewes (n = 3) and maintenance (M) nutrition ewes (n = 3). The timeline at the top of the diagram represents the gestational period. D0: day 0 of gestation; D55: days 55 of gestation; D85: days 85 of gestation; D105: days 105 of gestation; D135: days 135 of gestation; P7: postnatal day 7; P7: postnatal day 35. The solid line represents the maintenance nutrition feeding period, the dotted line represents the Sub-maintenance nutrition feeding period. Triangle represents sampling time points. The vertical dotted lines are used to align the sampling time points of the Sub-M and M.

**Figure 2 animals-10-01058-f002:**
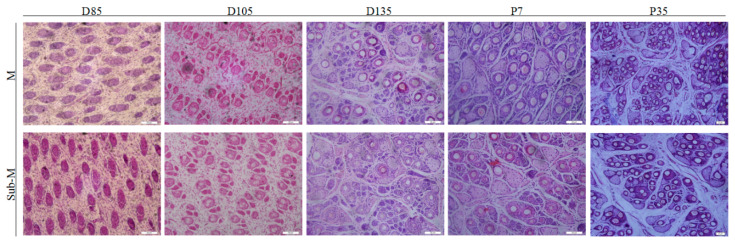
Morphogenesis and development of wool follicles in pregnancy maintenance and nutritional restriction group of Merino ewes. Lateral skin from fetal and lamb bodies was obtained; the deparaffinized sections were stained with H&E. Scale bars: 20 μm. The timeline at the top of the diagram represents the gestational period: D85, 85 days of gestation; D105, 105 days of gestation; D135, 135 days of gestation; P7, 7 days after birth; and P35, 35 days after birth.

**Figure 3 animals-10-01058-f003:**
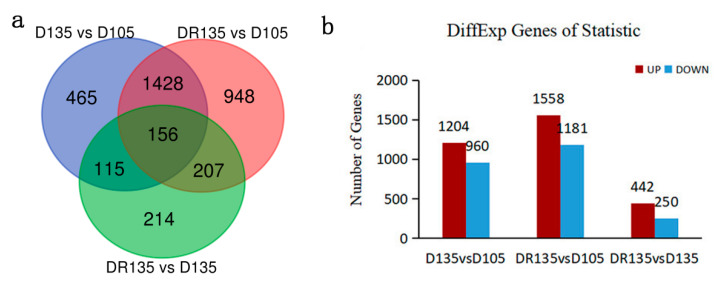
Summary of DEGs. (**a**) Venn diagram illustrating the overlaps of DEGs between D135 vs. D105, DR135 vs. D105 and DR135 vs. D135. (**b**) Number of up-regulated and down-regulated expression genes between D135 vs. D105, DR135 vs. D105, and DR135 vs. D135. The x-axis represents the comparison scheme between the indicated different time points. The y-axis represents the corresponding number of DEGs. The red columns represent up-regulated expression genes and blue columns represent down-regulated expression genes.

**Figure 4 animals-10-01058-f004:**
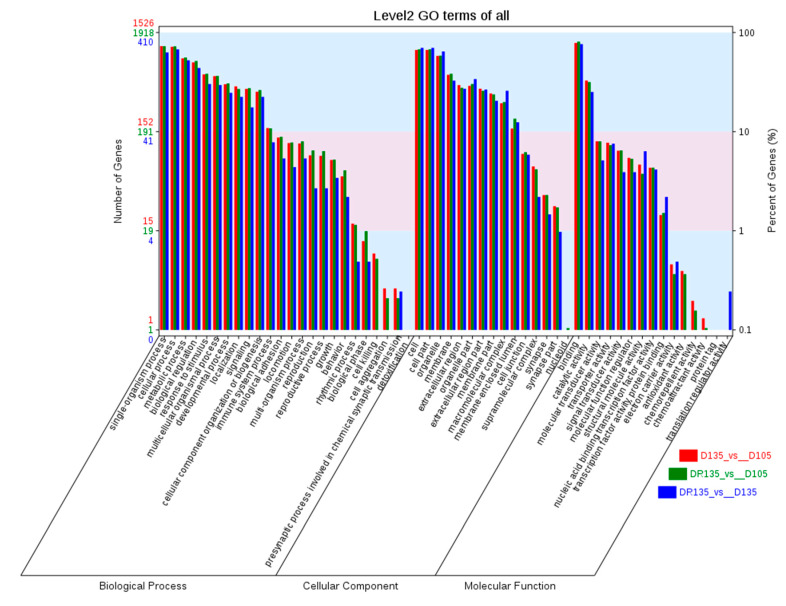
A Gene Ontology (GO) functional enrichment analysis of DEGs. The x-axis shows the three categories of Gene Ontology; the y-axis shows the number of genes (left) and percent of genes (right). The red bars represent DEGs number in D135 vs. D105. The green bars represent DEGs number inDR135 vs. D105 and Blue bars represent DEGs number inDR135 vs. D135.

**Figure 5 animals-10-01058-f005:**
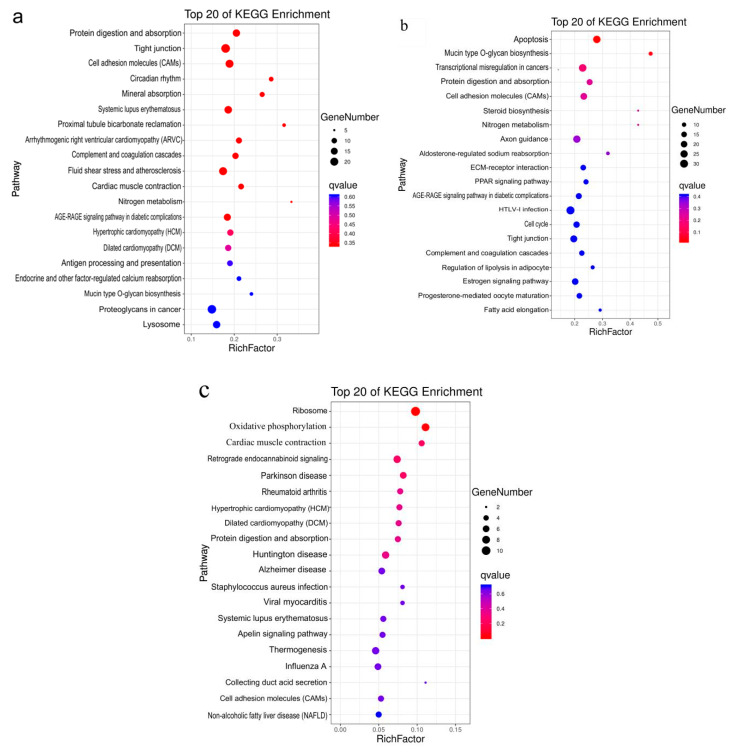
Top 20 KEGG enrichment pathways of DEGs. (**a**) KEGG enrichment of DEGs in D135 vs. D105. (**b**) KEGG enrichment of DEGs in DR135 vs. D105. (**c**) KEGG enrichment of DEGs in DR135 vs. D135. The x-axis represents rich factor and the y-axis represents the KEGG pathways.

**Figure 6 animals-10-01058-f006:**
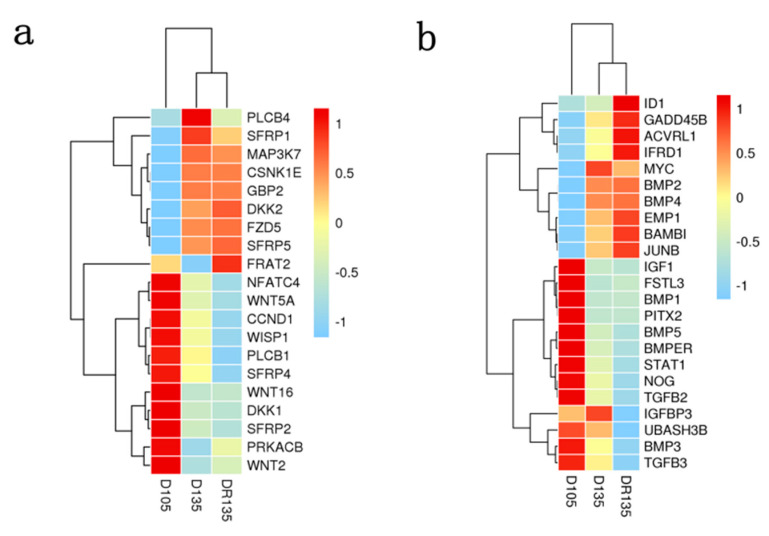
Heat map of the hierarchical clustering of DEGs. (**a**) Clustering heat map of Wnt signal pathway correlated genes. (**b**) Clustering heat map of TGF-β/BMP signaling pathway genes. Each row represents the DEGs, and each column represents samples at a time point. Heat map uses FPKM value and then standardizes. Blue and red gradients indicate a decrease and increase in transcript abundance, respectively. The line on the left represents gene cluster, and the line on the top represents sample cluster.

**Figure 7 animals-10-01058-f007:**
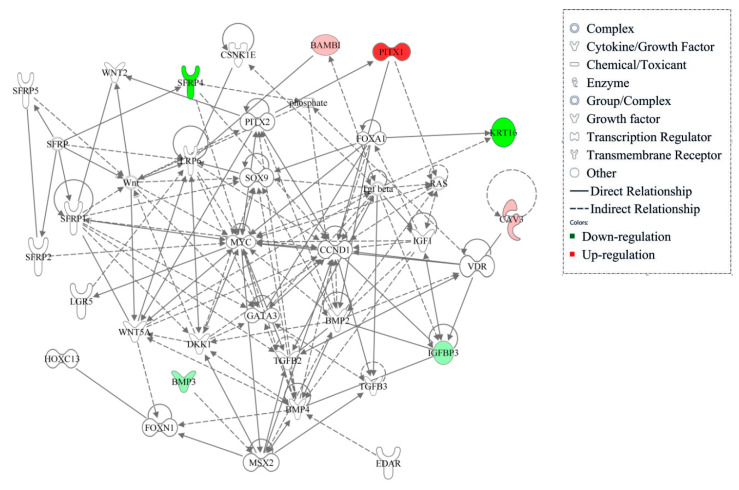
Interaction of potential genes related to wool follicles development. Color indicates the expression of the gene in DR135 vs. D135. Red represents up-regulation of gene expression and green represents down-regulation of gene expression (|log2 (fold change)| > 0.5). The intensity of green and red molecule colors indicates the degree of down or up-regulation, respectively. White molecules are not differential expression (|log2 (fold change)| < 0.5).

**Table 1 animals-10-01058-t001:** Wool follicles density and S/P ratios of M and Sub-M group at different gestation stages.

Time	Groups	Number	Primary Wool Follicles Density (mm^2^)	Secondary Wool Follicles Density(mm^2^)	Total Density (mm^2^)	Proportion of Secondary to Primary Wool Follicles
D85	M	3	84.15 ± 12.72	23.14 ± 14.10	107.30 ± 2.62	0.30 ± 0.13
	Sub-M	3	82.83 ± 34.18	30.66 ± 9.11	113.49 ± 43.24	0.31 ± 0.07
D105	M	3	71.75 ± 9.84	144.89 ± 34.17	216.64 ± 40.31	2.02 ± 0.41
	Sub-M	3	72.61 ± 9.25	114.14 ± 22.83	187.05 ± 14.01	1.60 ± 0.50
D135	M	3	28.56 ± 6.59	201.68 ± 33.89 ^a^	230.24 ± 39.97 ^a^	7.19 ± 0.71 ^a^
	Sub-M	3	26.99 ± 3.72	139.34 ± 17.30 ^b^	166.33 ± 19.86 ^b^	5.25 ± 0.60 ^b^
P7	M	3	13.29 ± 1.71	113.75 ± 22.98	127.04 ± 24.69	8.62 ± 0.75
	Sub-M	3	16.50 ± 3.19	154.50 ± 11.31	171.00 ± 14.50	9.59 ± 1.30
P35	M	3	6.67 ± 0.46	69.25 ± 11.19	91.53 ± 8.90	10.34 ± 1.03
	Sub-M	3	9.88 ± 1.94	95.95 ± 24.67	105.82 ± 26.62	9.79 ± 0.50

Data are means ± SD, ^a,b^ Within column, different superscript letter indicates differences (*p* < 0.05) between M and Sub-M group at the same gestation stages.

**Table 2 animals-10-01058-t002:** Potential key genes that are differentially expressed related to SF development.

Gene	Description	log2 (Fold Change)		
		D135 vs. D105	DR135 vs. D105	DR135 vs. D135
**Wnt related**				
*SFRP2*	Secreted Frizzled Related Protein 2	−2.40341	−2.79309	−0.38968
*SFRP4*	Secreted Frizzled Related Protein4	−1.07911	−2.10080	−1.02169
*SFRP5*	Secreted Frizzled Related Protein5	1.45166	1.62546	0.17380
*WNT5A*	Wnt Family Member 5A	−0.37882	−0.51687	−0.13805
*WNT2*	Wnt Family Member 2	−1.14818	−0.92800	0.22018
*DKK1*	Dickkopf-Related Protein 1	−1.44226	−1.56450	−0.12224
*CCND1*	Cyclin D1	−0.40895	−0.70606	−0.29711
*CSNK1E*	Casein Kinase 1 Epsilon	2.69495	2.67435	−0.02060
**TGF-β/BMP**				
*BMP2*	Bone Morphogenetic Protein 2	1.10571	1.19215	0.08644
*BMP3*	Bone Morphogenetic Protein 3	−0.57071	−1.11402	−0.54331
*BMP4*	Bone Morphogenetic Protein 4	0.78090	0.83946	0.05856
*BMP5*	Bone Morphogenetic Protein 5	−2.48384	−2.95107	−0.46723
*TGF-β2*	Transforming Growth Factor Beta 2	−0.73641	−1.11444	−0.37803
*TGF-β3*	Transforming Growth Factor Beta 3	−0.33778	−0.75208	−0.41430
*BAMBI*	BMP And Activin Membrane-Bound Inhibitor	1.05036	1.55454	0.50418
*MYC*	V-Myc Avian Myelocytomatosis Viral Oncogene Homolog	0.57669	0.42712	−0.14958
**Wool follicle** **development and transcriptional regulator genes**				
*PITX1*	Paired Like Homeodomain 1	−2.93483	−0.74502	2.18981
*CAV3*	Caveolin 3	−3.14244	−2.48249	0.65995
*SIX4*	SIX Homeobox 4	−2.38845	−1.90240	0.48605
*SLURP1*	Secreted LY6/PLAUR Domain Containing 1	4.00780	3.56769	−0.44011
*GATA3*	GATA Binding Protein 3	0.71200	0.40038	−0.31162
*EDAR*	Ectodysplasin A Receptor	−0.91182	−1.05320	−0.14137
*FOXN1*	Forkhead Box N1	2.21443	1.85361	−0.36083
*HOXC13*	Homeobox C13	0.69009	0.73500	0.04492
*LGR5*	Leucine-Rich Repeat Containing G Protein-Coupled Receptor 5	0.85120	0.84652	−0.00468
*MSX2*	Msh Homeobox 2	0.36489	0.65509	0.29020
*DNASE1L2*	Deoxyribonuclease 1 Like 2	1.55896	2.07115	0.51219
*VDR*	Vitamin D Receptor	1.10678	0.95640	−0.15039
*IGFBP3*	Insulin-Like Growth Factor Binding Protein 3	0.17093	−0.45424	−0.62518
**Branching morphology related genes**				
*PITX2*	Paired Like Homeodomain 2	−3.06724	−3.39049	−0.32325
*MMP12*	Matrix Metallopeptidase 12	3.74146	3.73938	−0.00208
*IGF1*	Insulin-Like Growth Factor 1	−0.71163	−0.75344	−0.04181
*SOX9*	SRY-Box 9	0.95372	0.63590	−0.31782
*FOXA1*	Forkhead Box A1	2.24977	2.56852	0.31876
**Keratin differentiation-related genes**				
*KRT16*	Keratin 16	1.27650	−1.44261	−2.71911
*KRTAP8-1*	Keratin Associated Protein 8-1	4.37327	4.93892	0.56566
*KRT13*	Keratin 13	−7.51172	−6.59676	0.91496
*KRT78*	Keratin 78	−2.72411	−2.20584	0.51827

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
