# Peer review of "Effect of Nutritional Restriction on the Hair Follicles Development and Skin Transcriptome of Chinese Merino Sheep"

_animals, 2020, doi:10.3390/ani10061058_

Round 1

Reviewer 1 Report

The authors agreed with most of my suggestions, and gave explanations for the rest.

The manuscript has been substantially improved and I believe that it can be published in its present form.

Author Response

Dear Reviewer:

Thank you very much for your approval of our manuscript.

Reviewer 2 Report

I believe that the manuscript was improved by the authors which provided answers to my questions and gave another viewpoint to the manuscript focusing on gene expression related to wool follicle morphogenesis. I have some questions/suggestions.

I cannot see the supplementary files, please use a standard file type (PDF, DOC, ZIP). 

Introduction

I liked the corrections you made, changing the focus of the study, but need some change.

Line74: The purpose of this study was to establish a method of different wool follicle development through nutritional restriction and to identify possible genes involved in the development of SF and secondary-derived wool follicles.

Please replace for: The purpose of this study was to identify possible genes involved in the development of SF and secondary-derived wool follicles and to evaluate the effects of sub-nutrition at different gestational moments at wool follicle and development.

There are important papers related to this theme that are missing and must be cited, please see:

https://doi.org/10.3390/ani9070450

https://doi.org/10.3390/ani9050214

https://doi.org/10.3390/ani9040142

https://doi.org/10.3390/ani10010033

The methods should be divided into animals and feed and insert a separate section about study design.

Regarding my previous comment concerning the number of animals. At line 80 they say” Chinese Merino ewes weighing 40 - 60 kg”. Why not inform the number of animals here. Figure 1 legend presents the following: “Timelines of the maternal feed restriction experiment. Feed restrictions are divided into five periods (1-5), each period included sub-maintenance (Sub-M) nutrition ewes (n=3 to 5) and maintenance (M) nutrition ewes (n=3 to 5).” I can understand how many sheep were used. 3 to 5 is not a number. Why the author just say how many animals in each group in each period? The study design, in my opinion, is still unclear. At table 1 I see 30 sheep, 3 per group, so be clear at the beginning of the methods section about how many animals did you used. 

Figure 1 must have one legend in the top left corner with the day. Maybe day 0. 

Methods: L86 “Based on previous studies of hair follicles morphogenesis and development[1,11], the five periods of sub-maintenance nutrition (Sub-M) were: from days 55 to 85, 88 85 to 105, and 105 to 135 of gestation, day 135 of gestation to 7 days after birth, and 7 to 35 days after 89 birth.

This is unclear. The author mentions the five periods of sub-maintenance but no mention before in the methods section about this. Be clear about your study: We developed a randomized factorial 2 x 5 design with 2 feed conditions (M and Sub-M) and 5 different physiological moments (or periods) (XX, YY etc) with 3 repetitions (Correct?).

Line 142: Hence, 3 independent skin samples from day 105 of gestation for the M group (D105), day 135 of gestation for the M group (D135) and D135 of gestation for the Sub-M group (DR135) were analyzed.

Why not samples from sub-M group at D105?? 

Results

At tables and figures, all the abbreviations must be defined.

Regarding my previous comment:

Comment: How the sub-nutrition after the birth would impact the wool morphogenesis since the authors claim that is embryonic? poor nutrition would impact milk production? 

Reply: Based on previous study (Rasmussen, P.V. 2000), nutrition condition of pregnancy playing a critical role for hair follicles morphogenesis and development had been demonstrated. Ewe malnutrition inhibited fetal hair follicle morphogenesis, development and reduced the milk production.

I think the authors don´t understand my comment. I understand the nutrition during pregnancy would impact hair follicles morphogenesis, my question was, how poor nutrition after birth would impact?? They do no answer this. Sub-M 5 group has poor nutrition after birth, so how the poor nutrition would affect? That was my question. What was the rationality in including a group with poor nutrition after birth, since the hair is developed during pregnancy? 

Figure 1 should have 2 lines, one representing the sub-M group and another line representing the M group, and with the number of sheep informed for each group. 

Conclusion

I think that some input regarding the differences of gene expression between M and sub-M group

Author Response

Dear Reviewer:

Thank you for the comments.Those comments are valuable and very helpful for revising and improving our paper. We have studied comments carefully and have made corretion which we hope meet with approval. Revised portion are marked in red in the paper.

Comment1:I cannot see the supplementary files, please use a standard file type (PDF, DOC, ZIP). 

Reply:Supplementary files have been submitted use *.doc type.

Comment2: Line74: The purpose of this study was to establish a method of different wool follicle development through nutritional restriction and to identify possible genes involved in the development of SF and secondary-derived wool follicles.

Please replace for: The purpose of this study was to identify possible genes involved in the development of SF and secondary-derived wool follicles and to evaluate the effects of sub-nutrition at different gestational moments at wool follicle and development.

Reply: Line74 has been revised according to reviewer's comments.

Comment3:There are important papers related to this theme that are missing and must be cited,

https://doi.org/10.3390/ani9070450 

https://doi.org/10.3390/ani9040142  

https://doi.org/10.3390/ani10010033

https://doi.org/10.3390/ani9050214 

Reply:These articles include candidate genes associated with wool production and traits, as well as function of keratin-associated proteins, which have been inserted into corresponding positions in the manuscript.

Comment4:The methods should be divided into animals and feed and insert a separate section about study design.

Reply:According to the reviewer's suggestion, L79 2.1: animal and study design has been divided into:2.1 study design and 2.2 animal feeding.

Comment5:Regarding my previous comment concerning the number of animals. At line 80 they say” Chinese Merino ewes weighing 40 - 60 kg”. Why not inform the number of animals here. Figure 1 legend presents the following: “Timelines of the maternal feed restriction experiment. Feed restrictions are divided into five periods (1-5), each period included sub-maintenance (Sub-M) nutrition ewes (n=3 to 5) and maintenance (M) nutrition ewes (n=3 to 5).” I can understand how many sheep were used. 3 to 5 is not a number. Why the author just say how many animals in each group in each period? The study design, in my opinion, is still unclear. At table 1 I see 30 sheep, 3 per group, so be clear at the beginning of the methods section about how many animals did you used. 

Figure 1 must have one legend in the top left corner with the day. Maybe day 0. 

Figure 1 should have 2 lines, one representing the sub-M group and another line representing the M group, and with the number of sheep informed for each group. 

Reply:According to the reviewer's suggestion,we modified study design and Figure 1, adding the number of samples for each group. and also see lines 88-97.

Comment6:Methods: L86 “Based on previous studies of hair follicles morphogenesis and development[1,11], the five periods of sub-maintenance nutrition (Sub-M) were: from days 55 to 85, 88 85 to 105, and 105 to 135 of gestation, day 135 of gestation to 7 days after birth, and 7 to 35 days after 89 birth.

This is unclear. The author mentions the five periods of sub-maintenance but no mention before in the methods section about this. Be clear about your study: We developed a randomized factorial 2 x 5 design with 2 feed conditions (M and Sub-M) and 5 different physiological moments (or periods) (XX, YY etc) with 3 repetitions (Correct?).

Reply:The primary hair follicle of fine wool sheep occurs at about day 55 of gestation, secondary wool follicles starting at approximately day 85 of gestation. The secondary-derived wool follicles initiated at approximately day 105 until day 135 of gestation. It is reported to complete the maturation process of secondary wool follicle in the first four to five weeks (28–35 days) after birth (Fraser,A.S. 1954), it is feasible that manipulation of the level of nutrition for a short period could have a marked effect on the adult follicle population.

   In earlier research, it was reported that maturation of the secondary follicle population could be influenced by adverse nutrition(Schinckel, P.G,1953).Therefore, we designed the experimental program according to the wool follicles morphogenesis and development stage of fine wool sheep.

We have revised lines 88-95.

Comment7:Line 142: Hence, 3 independent skin samples from day 105 of gestation for the M group (D105), day 135 of gestation for the M group (D135) and D135 of gestation for the Sub-M group (DR135) were analyzed.

Why not samples from sub-M group at D105?? 

Reply: Although there was no significant difference in the primary wool follicles and secondary wool follicles density between the sub-M group and the M group at days 105 of gestation, but the density of hair follicles in sub-M group was lower than that in M group. It indicated that nutrition restriction has begun to inhibit the development of wool follicles in sub-M group.In order to study the difference of wool follicle development between sub-M and M groups from 105-135 days of gestation, only 105 days of gestation samples of M group were selected. 

Comment8: At tables and figures, all the abbreviations must be defined.

Reply: Abbreviations in Table 1 and figure 1 have been revised.

Comment9: How the sub-nutrition after the birth would impact the wool morphogenesis since the authors claim that is embryonic? poor nutrition would impact milk production? 

I think the authors don´t understand my comment. I understand the nutrition during pregnancy would impact hair follicles morphogenesis, my question was, how poor nutrition after birth would impact?? They do no answer this. Sub-M 5 group has poor nutrition after birth, so how the poor nutrition would affect? That was my question. What was the rationality in including a group with poor nutrition after birth, since the hair is developed during pregnancy? 

 Reply:It is reported to complete the maturation process of secondary wool follicle it can take up  five weeks after birth(Fraser,A.S. 1954). Sub-maintenance nutrition will reduce the milk production of ewe after birth. In the study, the lambs were fed only the milk of the ewe within 35 days after birth without supplementary feeding. Reduced milk intake may affect the development of lamb wool follicle and delay the maturation of secondary hair follicles. We have found that the mature of secondary wool follicle of the lamb was delayed, and the growth of the hair fiber was inhibited, through the histologic analysis of the lamb wool follicle on day 7 and 35 after birth.

See lines 92-95 add in study design.

Reviewer 3 Report

Thank you for the answers. However, there is still need for some improvement.

Comment2:There was no validation step based on RT-PCR to confirm RNASeq results. Could you tell me why?

Reply: In this study, three different individuals were selected for each developmental stage, and the three RNA SEQ sequencing data were set as the biological repeat of the same period. The square of Pearson correlation coefficient (R2) of sequencing sequence of three samples in the same period was greater than 0.9. It shows that the expression patterns of samples from different individuals in the same period are basically the same, and the repeatability is very high. Therefore, we think that the sequencing results can reflect the trend of gene change, so we did not put the PT-PCR results into the paper.

In fact, we chose some genes to complete the RT-PCR experiment, which showed the consistence with RNASeq results. However, the result don't exhibited in our manuscript in consideration of the requirement of other research. This manuscript focused on the analysis of RNASeq. The data of this manuscript was sufficient.

The authors raised two different tasks:

  • The square of Pearson correlation coefficient (R2) of sequencing sequence” How you used this coefficient of determination for sequences? Do you mean Pearson correlation coefficient for determined counts of three samples? Is not mean that the study is repeatable in wide-scale, it more means that your probes, libraries and methodology was good. Still, the results could not be transferred to a wide scale.
  • the result don't exhibited in our manuscript in consideration of the requirement of other research” The validation based on RT-PCR is an integral part of RNASeq study to verified their repeatability. I strongly suggest adding this part to the manuscript. However, the final decision I will leave to the editor.

Comment3: The raw dataset was not submitted to the public database. Could you explain this?

Reply: We planned to upload the raw dataset to SRA, if that was required by the magazine.

I strongly suggest uploading the raw dataset for some public database (SRA/GEO). However, the final decision I will leave to the editor.

Comment6: There were no full lists of DEGs as additional files.

Reply: According to the reviewer's suggestion, the DEG list as supplementary materials (Table 4-6) have been submitted.

I strongly suggest uploading the DEGs in more usable form (csv or excel file), not in pdf. I found new transcripts described as Novel**** or TCONS****. Did the authors try to do some BLAST for this sequences? It is good to describe more about this situation.

Additionally, the fist columns correspond to transcript_id (ENSOART00000019247), not to the gene (DCX). Is the terminology should be uniform and transparent.

Comment9: Why was the |log2 (fold change)| higher than 0.5 selected? The 0.5 looks like a shallow threshold, and this could provide a significant number of up and down-regulated genes.

Reply: The number of differentially expressed genes will decreased when the |log2 (fold change)| higher than 1.0, and we found many genes related to hair follicle development were filtered out, such as BMP3、BMP4、IGF1、KRT80、WNT16、DKK2、HOXC13、TGF-β2、TGF-β2. In order to obtain more comprehensively data, |log2 (fold change)|>0.5 were choose as the criteria to filter the DEGs.

Is that mean that the authors focused more on well-known genes/transcripts, not for new one identified in the manuscript.

Author Response

Dear Reviewer:

Thank you for the comments,those comments are valuable and very helpful for revising and improving our paper, as well as the important guiding significance to our researches.we have studied comments carefully and have made corretion which we hope meet with approval. Revised portion are marked in red in the paper. 

Comment1:There was no validation step based on RT-PCR to confirm RNASeq results. Could you tell me why?

Reply: we chose some genes to complete the RT-PCR experiment, which showed the consistence with RNASeq results. However, the result don't exhibited in our manuscript in consideration of the requirement of other research.

We added the qRT-PCR verification experiment to the manuscript(see lines174-180 and 306-310),the results of qRT-PCR verification were submitted in the supplementary files. 

Comment2: The raw dataset was not submitted to the public database. Could you explain this?

Reply: Transcription data will be further studied and analyzed in our current study. Then we will upload the data to public database, if that was required by the journal.

Comment3: I strongly suggest uploading the DEGs in more usable form (csv or excel file), not in pdf. I found new transcripts described as Novel**** or TCONS****. Did the authors try to do some BLAST for this sequences? It is good to describe more about this situation.

Additionally, the fist columns correspond to transcript_id (ENSOART00000019247), not to the gene (DCX). Is the terminology should be uniform and transparent.

Reply: According to the reviewer's suggestion, the list of differential genes has been revised and submitted in a excel file. Thank you very much for your suggestion, we will focus on new transcription / genes in further research.

In the DEGs list, the first columns correspond to transcript_id, and the second column correspond to gene_id.

Comment4: Why was the |log2 (fold change)| higher than 0.5 selected? The 0.5 looks like a shallow threshold, and this could provide a significant number of up and down-regulated genes.

Reply: A total of 2,164, 2,739, and 692 DEGs in D135 vs D105, DR135 vs D105, and DR135 vs D135, respectively, were obtained. If the|log2 (fold change)|>1, the number of differentially expressed gene are 837, 960 and 160 after annotation, respectively.To some extent, it has influence on the analysis of KEGG enrichment and IPA network analysis. In fact, key genes with |log2 (fold change)|>1 were selected in our study.

Thank you very much for your comments. We will pay attention to the new transcripts/genes in further research.

Round 2

Reviewer 2 Report

The authors revise the manuscript according to my comments and I am satisfied with their comments.

I have one minor suggestion, in figure 1 you don´t need to repeat the number of animals for each group in the figure it self. I made this suggestion because in the first version of this manuscript you inform 2 or 3 animals per group. If all groups have the same number of animals just say in the text before the figure or in the figure legend.

Author Response

Dear Reviewer:

Thank you for you suggestion.

We deleted the number of animals in Figure 1.In 2.1 Study design, number of animals were mentioned. “the effects of nutrition in five time periods were examined.These included days 55 to 85, 85 to 105, and 105 to 135 of gestation, day 135 of gestation to 7 days after birth, and 7 to 35 days after birth. During each period ewes were offered either sub-maintence or maintenance feeding with 3 ewes in each subgroup”.

The number of experimental animals were also mentioned in Figure legend .“The experiment are divided into five groups (1-5), each group included sub-maintenance (Sub-M) nutrition ewes (n=3) and maintenance (M) nutrition ewes (n=3)”.

Reviewer 3 Report

I am accepting paper in present form.

Author Response

Dear Reviewer:

Thank you very much for your approval of our manuscript.We checked the manuscript carefully again and corrected some minor spell mistakes.

This manuscript is a resubmission of an earlier submission. The following is a list of the peer review reports and author responses from that submission.

Round 1

Reviewer 1 Report

The authors presented an interesting paper on studies of the effect of the restricted nutrition on development of the hair follicles and skin transcriptome of Chinese Merino sheep.

The prenatal and postnatal development of wool follicles, as well as the transcriptional expression profile of fetus’ and new-born lambs’ skin was characterized under normal and restricted maternal nutrition.

The present work is a useful study; however, in some parts, the manuscript is difficult for understanding. Sometimes, I had to read sentences two – three times to understand, what the authors mean. The text of the manuscript should be significantly revised (see below) before it will be accepted for publication. The English language must be significantly improved.

The title of the work corresponds to the content of the research. The abstract summarises the main points of the work. Introduction section gives the short description about state of the art of the research area.

The aim of the research is defined not clearly:

P2 L70 … we collected fetal tissue at 5 different time points…

Only three time points correspond to the embryonal period, while the 4th and 5th time points correspond to the 7th and 35th days after birth. Thus, it is not correct to write “fetal tissue”.

P2 L72 please, decipher the abbreviation “DGE”

The methodology is complete, but some clarifications are needed. I suggest to rephrase the description of groups (P2 L89 – P3 L 96) to make it more clearly for understanding.

The results are completely presented but some points should be clarified.

At description of results the authors mainly use the abbreviation D55, D85 etc. and do not specify the groups (Sub-M1, Sub-M2 etc.). I suggest to specify the group through the text at description of research results.

P4 L 134 “between M and Sub-M group at days 135 of gestation”. Please, specify, which Sub-M group do you mean (Sub-M3?)  

P4 L135 please, correct, “and day 105 135 of gestation (D135) for M group and D135 of gestation for Sub-M (Sub_m3?) group (DR135). What is DR135?

P4 L139-140 “representing samples from the time points at which wool follicles densities were significantly”. Please, specify the time points.  

P5 L185 Figure 2. Please, indicate the group number below the pictures.  

P6 L202 Please, correct. The transcriptional profiles of D105, D135 and DR135 samples identified 20062, 20653 and 20846 transcripts in D105, D135 and DR135, respectively

There are a lot of inaccuracies in the text of the article. I didn't correct the text further. I recommend that the authors carefully go through the text and correct any inaccuracies. I recommend improving the English language and submitting the revised manuscript for review.

Reviewer 2 Report

The manuscript entitled "Effect of nutritional restriction on the hair follicles development and skin transcriptome of Chinese Merino sheep" shows that poor food impairs wool development. I don´t see what is the scientific information valid obtained from this work to justify this kind of experiment with the cesarian of several sheep with no mention to animal ethics approval. There are several typos in the manuscript and gross errors (e.g. methodology in the introduction, results in the methodology). The study design is not clear at all. They don´t mention the number of sheep sampled. They say that after results from D135 they decided to sample for transcriptome sequencing on day 105. How this would be possible (maybe by scaling, or pilot study, but those are not mentioned). No mention again about the number of animals. No raw data present, which is mandatory for this type of manuscript. How the sub-nutrition after the birth would impact the wool morphogenesis since the authors claim that is embryonic? poor nutrition would impact milk production? very poor design and I don´t think this manuscript is valid for publication in the journal.
Other comments:
L166: After feed restriction, most Blood Biochemical indexes were no difference in the M group and Sub-M group
Where is the data? I could not find table S1 or S2 in the submission system. Very poor English. The statistical analysis considered each time separately but it should be performed as a repeated measures test.
L425: Sub-M nutrition (75% maintenance nutrition) has no significant effect on ewes and fetal development.
How they reach such a conclusion? is not supported by the data.

L428: In this study, we developed a method to study the prenatal and postnatal development of the wool follicles under pregnancy maintenance feeding and Sub nutrition in Merino sheep
How this "model" will be important for science? I don´t see the point to develop such a model and the design is not clear and can not be defined as a model.

Reviewer 3 Report

Comments:

  1. The authors used “Based on previous studies of wool follicles morphogenesis and development…” (line 91). There was no citation or comprehensive description for what the paragraph was supported. Could you explain this?
  2. There was no validation step based on RT-PCR to confirm RNASeq results. Could you tell me why?
  3. The raw dataset was not submitted to the public database. Could you explain this?
  4. The research is not based on ethics statement. Could you tell me why?
  5. What version of bioinformatic software was used?
  6. There were no full lists of DEGs as additional files.
  7. One time the authors used adjusted p-value, one q-value. The terminology should be same or explained in M&M.
  8. What software was used to count reads?
  9. Why was the |log2 (fold change)| higher than 0.5 selected? The 0.5 looks like a shallow threshold, and this could provide a significant number of up and down-regulated genes.
  10. Why were the clean reads aligned to the genome and not to transcriptome? Could you show the alignment and transcriptome statistics, also with the MDS/PCA plots?
  11. Why did the authors decide to used RPKMs instead of raw counts to estimate DEGs?
  12. The authors used on RPKMs but too creates plots (Figure 6) FPKM. What was the reason to used two different count measures and how the authors estimate them?
  13. Is there was any of the comprised groups (D105-D135, D105-DR135, D135-DR135) assigned to only one category of GO or KEGG? Based on Figure 4, the groups were just as numerous. Did you do some statistics (Tukey? Or another test) on that?
  14. Is it possible to add some time changes on plot like Sankey diagram https://www.r-graph-gallery.com/sankey-diagram.html and combine Figure 5 into one diagram? Or for interesting genes (Figure 6)? And Figure 7 – co-expression/interaction?
  15. The improvement in English is needed. Also based on the composition of the text (double spaces, parentheses & commas with and without spaces and so on).